

# Therapeutic targeting of ARID1A and PI3K/AKT pathway alterations in cholangiocarcinoma

Supharada Tessiri[1,2], Anchalee Techasen[1,3], Sarinya Kongpetch[3,4], Achira Namjan[1,2], Watcharin Loilome[3,5], Waraporn Chan-on[6], Raynoo Thanan[5] and Apinya Jusakul[1,3]

[1] Centre for Research and Development of Medical Diagnostic Laboratories, Faculty of Associated Medical Sciences, Khon Kaen University, Khon Kaen, Thailand
[2] Biomedical Science Program, Graduate School, Khon Kaen University, Khon Kaen, Thailand
[3] Cholangiocarcinoma Research Institute, Faculty of Medicine, Khon Kaen University, Khon Kaen, Thailand
[4] Department of Pharmacology, Faculty of Medicine, Khon Kaen University, Khon Kaen, Thailand
[5] Department of Biochemistry, Faculty of Medicine, Khon Kaen University, Khon Kaen, Thailand
[6] Center for Research and Innovation, Faculty of Medical Technology, Mahidol University, Bangkok, Thailand

Corresponding author
Apinya Jusakul, apinjus@kku.ac.th

## ABSTRACT

**Background.** Genetic alterations in *ARID1A* were detected at a high frequency in cholangiocarcinoma (CCA). Growing evidence indicates that the loss of ARID1A expression leads to activation of the PI3K/AKT pathway and increasing sensitivity of ARID1A-deficient cells for treatment with the PI3K/AKT inhibitor. Therefore, we investigated the association between genetic alterations of *ARID1A* and the PI3K/AKT pathway and evaluated the effect of AKT inhibition on ARID1A-deficient CCA cells.

**Methods.** Alterations of *ARID1A*, PI3K/AKT pathway-related genes, clinicopathological data and overall survival of 795 CCA patients were retrieved from cBio Cancer Genomics Portal (cBioPortal) databases. The association between genetic alterations and clinical data were analyzed. The effect of the AKT inhibitor (MK-2206) on ARID1A-deficient CCA cell lines and stable *ARID1A*-knockdown cell lines was investigated. Cell viability, apoptosis, and expression of AKT signaling were analyzed using an MTT assay, flow cytometry, and Western blots, respectively.

**Results.** The analysis of a total of 795 CCA samples revealed that *ARID1A* alterations significantly co-occurred with mutations of *EPHA2* ($p < 0.001$), *PIK3CA* ($p = 0.047$), and *LAMA1* ($p = 0.024$). Among the *EPHA2* mutant CCA tumors, 82% of *EPHA2* mutant tumors co-occurred with *ARID1A* truncating mutations. CCA tumors with *ARID1A* and *EPHA2* mutations correlated with better survival compared to tumors with *ARID1A* mutations alone. We detected that 30% of patients with *PIK3CA* driver missense mutations harbored *ARID1A*-truncated mutations and 60% of *LAMA1*-mutated CCA co-occurred with truncating mutations of *ARID1A*. Interestingly, ARID1A-deficient CCA cell lines and *ARID1A*-knockdown CCA cells led to increased sensitivity to treatment with MK-2206 compared to the control. Treatment with MK-2206 induced apoptosis in *ARID1A*-knockdown KKU-213A and HUCCT1 cell lines and decreased the expression of pAKT[S473] and mTOR.

**Conclusion.** These findings suggest a dependency of ARID1A-deficient CCA tumors with the activation of the PI3K/AKT-pathway, and that they may be more vulnerable to selective AKT pathway inhibitors which can be used therapeutically.

# INTRODUCTION

Cholangiocarcinoma (CCA) is a malignancy arising from the epithelial cells along the biliary tree. The highest incidence rates of CCA have been reported in northeast Thailand, which is the endemic area of the group 1 carcinogen, the liver fluke, *Opisthorchis viverrini* (Ov) (*Sripa & Pairojkul, 2008*; *Alsaleh et al., 2018*). Cholangiocarcinogenesis is induced via multifactorial mechanistic pathways. DNA damage and genetic alterations occur during CCA progression (*Sripa et al., 2007*; *Sripa et al., 2012*). Although the best treatment option for localized CCA is curative surgical resection the five-year survival rate after surgical resection is low approximately 30% to 60% (*Meza-Junco et al., 2010*; *Patel, 2011*). Thus, more treatment options for CCA patients are urgently needed. Growing evidence from molecular genetic studies of CCA has initiated a significant shift towards a precision medicine-based approach. In recent years, molecular targets with clinical significance include fibroblast growth factor receptor (*FGFR*), isocitrate dehydrogenase (*IDH1/2*), human epidermal growth factor receptor (*HER*), neurotropic tyrosine kinase receptor (*NTRK*) fusions, and *BRAF* mutations have been identified in CCA tumors (*Lamarca et al., 2020*).

AT-rich interactive domain containing protein 1A (ARID1A or BAF250a) is a crucial non-catalytic subunit of human switch/sucrose nonfermentable (SWI/SNF) complex (*Wu & Roberts, 2013*; *Xu & Tang, 2021*). It plays an important role in crucial cellular processes including transcription, DNA replication, and DNA damage repair (*Basu et al., 2016*; *Bayona-Feliu et al., 2021*). Of note, *ARID1A* is commonly inactivated in tumors including CCA (*Wu & Roberts, 2013*; *Jusakul et al., 2017*; *Orlando et al., 2019*). Silencing of *ARID1A* has resulted in a significant increase in proliferation *in vitro* (*Samartzis et al., 2014*). In CCA tumors, decreased expression of ARID1A was associated with CCA progression and metastasis, indicating the tumor-suppressor function of ARID1A in CCA (*Chan-on et al., 2013*; *Namjan et al., 2020*; *Zhao et al., 2021*). Recent studies have shown that *ARID1A* mutations is involved in carcinogenesis via activation of the phosphatidylinositol 3-kinase/protein kinase B (PI3K/AKT) pathway (*Takeda et al., 2016*). Co-occurrence of *ARID1A* alterations with PI3K/AKT pathway activation has been reported in ovarian clear cell carcinoma, breast cancer, and gastric cancer (*Huang et al., 2014*; *Samartzis et al., 2014*; *Zhang et al., 2016*; *De & Dey, 2019*). Silencing of *ARID1A* in gastric, ovarian, glioma, and colon cancer cells has been shown to activate the phosphorylation of AKT, and PI3K (*Zeng et al., 2013*; *Xie et al., 2014*; *Takeda et al., 2016*; *Zhang et al., 2016*), suggesting an interrelationship between *ARID1A* deficiency and PI3K/AKT pathway activation. Taken together, the crucial tumor suppressive roles of ARID1A shed light on targeted therapeutic strategies, hence there has been ongoing effort towards developing effective therapeutic strategies for *ARID1A* deficient tumors (*Bitler, Fatkhutdinov & Zhang, 2015*; *Mathur, 2018*; *Mullen et al., 2021*).

Interestingly, *ARID1A* mutations and the depletion of ARID1A protein expression sensitized cancer cells to PI3K/AKT inhibitors. In breast and gastric cancer, *ARID1A*-depleted cells showed an increased sensitivity to PI3K and AKT inhibitor compared to wild-type cells (*Samartzis et al., 2014*; *Zhang et al., 2016*; *Yang et al., 2018b*). This is of significant clinical importance since *ARID1A* mutations or loss of the expression can be predictive of a favorable therapeutic response to inhibitors in the PI3K/AKT pathway. Although *ARID1A* inactivation and PI3K/AKT pathway alteration frequently occur in CCA, the effect of PI3K or AKT inhibitor has not been well-defined in *ARID1A*-deficient CCA. We therefore aimed to study the association between genetic alterations of *ARID1A* and the PI3K/AKT pathway in CCA and investigate the effect of AKT inhibitor on *ARID1A*-deficient CCA cell lines. This study will provide a unique opportunity for predicting favorable treatment responses to inhibitors of the PI3K/AKT pathway on *ARID1A*-deficient CCA tumors which might further improve treatment outcome.

## MATERIAL AND METHODS

### Cell lines and cell culture

The human cholangiocarcinoma cell lines KKU-452 (JCRB1772) (*Saensa-ard et al., 2017*), KKU-055 (JCRB1551), KKU-213A (JCRB1557) (*Sripa et al., 2020*), and KKU-100 (JCRB1568) (*Sripa, 2005*) were developed at Cholangiocarcinoma Research Institute, Khon Kaen University and deposited to the Japanese Cancer Research Resources Bank (JCRB, Ibaraki city, Osaka, Japan). The HUCCT1 (RCB1960) cell line was obtained from the RIKEN Bioresource Research Center (Ibaraki, Japan). The KKU-452, KKU-055, KKU-213A and KKU-100 cell lines were cultured in Ham's F-12 whereas the HUCCT1 cells were cultured in RPMI containing 10% fetal bovine serum and penicillin/streptomycin (100 U/ml and 100 μl/ml). The cells were incubated in a humidified incubator at 37 °C and 5% $CO_2$.

### Materials

The following reagents and antibodies were used: MK-2206 (A-1909: Active Biochem, Hong Kong) was dissolved in DMSO at a concentration of 10 mM as a stock solution, ARID1A (HPA005456; Sigma-Aldrich, Germany), phospho-AKT$^{S473}$ (pAKT$^{S473}$, SAB4300042; Sigma-Aldrich, Germany), AKT (#9272; Cell Signaling Technology, USA), mTOR (#2983; Cell Signaling Technology, USA), Bax (50599-2-Ig; Proteintech, USA), Bcl-2 (12789-1-AP; Proteintech, USA), and β-actin (A5441; Sigma-Aldrich, Germany).

### Analysis of gene alterations, using the open-access bio-database cBioPortal

We utilized the cBioPortal for Cancer Genomics (http://cbioportal.org), a web-based, open-access resource for the analysis of cancer genomics data from The Cancer Genome Atlas (TCGA) and The International Cancer Genome Consortium (ICGC). In the present study, somatic mutations of *ARID1A*, genes in RAS/PI3K/AKT pathways (mutation frequency ≥ 1.7% including: *TP53, ARID1A, KRAS, EPHA2, STK11, PIK3CA, RASA1, LAMA2, ERBB2, LAMA1, BRAF, ERBB4, FGFR2, PIK3R1, PTEN, KDR, NRAS,* and *TNN*), clinicopathological data and patient survival of 795 CCA patients/798 samples were

analyzed. Collectively, the six data sets included a TCGA data portal (Firehose Legacy) and a ICGC data portal (*Jusakul et al., 2017*; *Chan-on et al., 2013*; *Ong et al., 2012*; *Jiao et al., 2013*; *Lowery et al., 2018*) (Table S1). We utilized cBioPortal to analyze genetic alterations, co-occurrence, and mutual exclusivity in CCA tumors. The OncoPrint, co-occurrence and mutual exclusivity of gene mutations were applied according to the online instructions of the cBioPortal. The statistical test for detecting co-occurrence and mutual exclusivity were based on a one-sided Fisher Exact Test, and Benjamini–Hochberg FDR correction in 153 pairs of the 18 genes (Table S2).

## Stable shRNA transduction

The cholangiocarcinoma cell lines were infected with MISSION® Lentiviral Transduction Particles Clone containing specific short-hairpin RNA against ARID1A#1 (SHCLNV-NM_006015-TRCN0000059091; Sigma-Aldrich, Germany), ARID1A#2 (SHCLNV-NM_006015-TRCN0000059089; Sigma-Aldrich, Germany) and non-targeted shRNA (SHC016V; Sigma-Aldrich, Germany) using polybrene (EMD Millipore; Sigma-Aldrich, Germany). The following shRNA sequences were used: non-targeted shRNA: sense: CCGGGCGCGATAGCGCTAATAATTTCTC, shARID1A#1: CCGTTGATGAACT-CATTGGTT, and shARID1A#2: GCCTGATCTATCTGGTTCAAT.

After 24-hours of infection, cells were selected using 1–2 $\mu$g/ml of puromycin (Sigma-Aldrich, Germany). Expression levels of ARID1A were confirmed by real time-PCR and Western blot.

## Cell viability assay

Cell viability was determined by a methylthiazolyldiphenyl-tetrazolium bromide (MTT) assay (PanReac Applichem, Germany). Cells were seeded into 96-well plates (Corning, NY, USA). After 24-hour exposure of inhibitor, 100 $\mu$l of 0.5 mg/ml MTT reagent was added and incubated for 2 h. After adding DMSO for 15 min, the absorbance was measured using a microplate reader at a wavelength of 570 nm. Each experiment was performed in triplicate and the results were given as means $\pm$ SD. The percentage of cell viability was calculated using the formula: % cell viability= (Nt/Nc) x100. Nt and Nc refer to the absorbance of the treated and control groups, respectively.

## Western blot

Cells were lysed in RIPA lysis buffer. Protein lysates were centrifuge at 14,000g for 20 mins at 4 °C. Protein concentration was determined using the Pierce[TM] BCA Protein Assay Kit (Pierce Biotechnology, USA). Protein lysates were resolved by SDS-PAGE and transferred onto PVDF membranes. The membranes were blocked with 5% skim milk or BSA in 1xTBS (1M Tris HCl pH 7.4, 5M NaCl) for 1 h at room temperature. Membranes were subsequently incubated with primary antibodies overnight at 4 °C. After washing, the secondary goat anti-Rabbit IgG-HRP (G21234; Invitrogen, USA) or rabbit anti-Mouse IgG-HRP (A16166; Thermo Fisher Scientific, USA) was used. The immunoreactive signals were visualized using Amersham[TM] ECL[TM] Prime Western Blotting Detection Reagent (GE Healthcare, UK).

## Apoptosis assay

Cell apoptosis was detected using an Annexin V-FITC and propidium iodide (PI) Kit (V13241; Invitrogen, USA) according to the manufacturer's protocol. Briefly, cells were seeded into 6-well culture plates overnight. Cells were then exposed to MK-2206 at designated concentrations and 0.3% DMSO was used as the control. After 24 h, cells were trypsinized, washed with ice-cold PBS, and resuspended in binding buffer containing Annexin V-FITC, whereupon, Annexin V/PI was added. Cells were resuspended in reaction buffer containing PI and immediately analyzed by BD FACSCanto II Flow cytometry (Becton Dickinson, USA) to detect the rate of apoptosis.

## RNA extraction and real time-RT PCR

Total RNA was isolated from cell pellets using TRIzol® Reagent (Invitrogen, USA) according to the manufacturer's protocol. Subsequently, 2 μg of total RNA was converted to cDNA using High-Capacity cDNA Reverse Transcription Kit (Applied Biosystems, USA). Real-time reverse transcription polymerase chain reaction (real time PCR) was performed using TaqMan gene expression assay: TaqMan probes (Hs00195664_m1 ARID1A and hs99999903_m1 β-actin; ThermoFisher Scientific, USA) to detected mRNA levels of ARID1A and β-actin. Real time PCR was performed using the ABI real-time PCR system, Quantstudio™ 6 Flex (Life technologies, Singapore). β-actin was used as the housekeeping gene.

## Statistical analysis

All experiments were repeated at least two times. Data were expressed as the mean ± standard deviation (SD). Statistical analysis was performed using SPSS 23.0 software (SPSS Inc., USA) or GraphPad Prism v.8.0 (GraphPad Inc., La Jolla, CA, USA) software. Overall survival (OS) curves were constructed according to the Kaplan–Meier estimator and differences between curves were tested for significance by means of log-rank tests. The half inhibitory concentration ($IC_{50}$) values were calculated by dose–response curves ($Y = 100/(1 + X/IC_{50})$). A two-tailed unpaired $t$-test was used for two-group comparisons. For multiple group comparisons, one-way analysis of variance, Kruskal-Wallis test, and Fisher's Least Significant Difference Test (LSD Test) test were used. A two-sided $p<0.05$ was considered statistically significant. Adjusted $p$-values were calculated using Benjamini–Hochberg correction.

# RESULTS

## *ARID1A* mutations and their co-occurrence with alterations in PI3K/AKT pathway

To address if inactivation of *ARID1A* in CCA was associated with alterations of PI3K/AKT signaling, the mutations of *ARID1A* and genes in RAS/PI3K/AKT pathway were assessed using cBioPortal. A total of 795 CCA patients were included in the present study. Among kinase-related genes, we selected 17 genes that were mutated (mutation frequency ≥1.7%) in CCA (Fig. S1). Somatic mutations of *ARID1A* were found in 19% (137/711) of CCA patients. Truncating *ARID1A* mutations were the most prevalent in *ARID1A* mutated CCA

(85%, 116/137) (Figs. 1–2, 3A). The most common truncating *ARID1A* mutation observed in this cohort was frameshift deletion (G276Efs*87, G277Rfs*123 and M274Ifs*126, Fig. 2). Additionally, truncating *ARID1A* mutations were more common in advanced stage (stage II (18%, 20/112), III (13%, 12/93) and IV (20%, 31/153), Fig. 3C). Suggesting that *ARID1A* mutation were predominantly in CCA with high tumor stage and may involve in CCA progression. We then investigated if *ARID1A* mutations co-occurred with mutations in the kinase-related pathway (Fig. S1). Interestingly, *ARID1A* mutations were significantly correlated with mutations of *EPHA2* (adjusted $p<0.001$), *PIK3CA* (adjusted $p = 0.047$), and *LAMA1* (adjusted $p = 0.024$) (Table S2). *EPHA2* was mutated in 9% (46/516) of CCA patients (Figs. 1 and 3A–3B). Truncating *EPHA2* mutations were the most prevalent alterations in *EPHA2* mutated CCA, particularly frameshift deletion (P460Rfs*33) (Figs. 1–2, 3A). Among *EPHA2* mutant CCA tumors, 48% (22/46) harbored *ARID1A* mutations (Figs. 1, 3B). Of note, 82% (18/22) of *EPHA2* mutant tumors co-occurred with *ARID1A* truncating mutations (Fig. 3B, Table S1). The frequency of *PIK3CA* mutations was 6% (41/711), and 80% (33/41) of *PIK3CA*-mutated CCA tumors were driver missense mutations (T1025A, E365K, E545K, E542K, C420R, Q546K, C901F, R38C, N345K, R88Q, R108H, M1043I, K111E, H1047R and H1047L) (Figs. 1–2, 3A and Table S1). We found that 37% (15/41) of CCA with *PIK3CA* mutations harbored *ARID1A* mutations. Of note, 30% (10/33) of CCA with *PIK3CA* driver missense mutations harbored *ARID1A*-truncated mutations (Fig. 3B, Table S1). *LAMA1* was mutated in 4% (20/516) of CCA (Fig. 1). *LAMA1* missense mutations were the most common mutations in CCA tumors (Figs. 2, 3A) and 50% (10/20) of *LAMA1*-mutated CCA co-occurring with *ARID1A* mutations (Fig. 1, Fig. 3B). Interestingly, 60% (6/10) of *LAMA1*-mutated CCA co-occurred with truncating mutations of *ARID1A* (Fig. 3B, Table S1). We did not detect any significant correlation between mutations in *ARID1A* and *KRAS*, *BRAF*, and *NRAS*. These results suggest interdependency between *ARID1A* mutations and alterations in the PI3K/AKT pathway, which may lead to activation of the PI3K/AKT pathway (Fig. S1).

## Coexistent *ARID1A-EPHA2* mutations associated with shorter overall survival in CCA patients

To examine the prognostic survival values of mutational co-occurrence between mutations of *ARID1A*, *EPHA2*, *PIK3CA* and *LAMA1* in CCA, we further performed survival analysis in CCA tumors with or without mutations of *ARID1A*, *EPHA2*, *PIK3CA* and *LAMA1* (Table S1). As shown in Fig. 4A, there was no significant correlation between *ARID1A* mutations and overall survival ($p = 0.190$, log-rank test), while co-occurrence of *ARID1A-EPHA2* mutations was significantly correlated with poor overall survival (HR = 2.651; 95% CI [1.34–6.19]; $p < 0.001$, log-rank test, Fig. 4B). In addition, patients with *ARID1A-EPHA2* mutations were found to have a shorter overall survival compared to patients with *ARID1A* mutations (HR = 1.905; 95% CI [0.92–3.95]; $p = 0.024$, log-rank test, Fig. 4C). In contrast, coexistent *ARID1A-PIK3CA* mutations and *ARID1A-LAMA1* mutations was not significantly associated with shorter overall survival (HR = 1.122; 95% CI [0.50–2.54]; $p = 0.768$ and HR = 0.598; 95% CI [0.25–1.66]; $p = 0.371$, respectively, log-rank test, Figs. 4D–4E). The insignificant shortening survival rate of the *ARID1A-PIK3CA* mutations

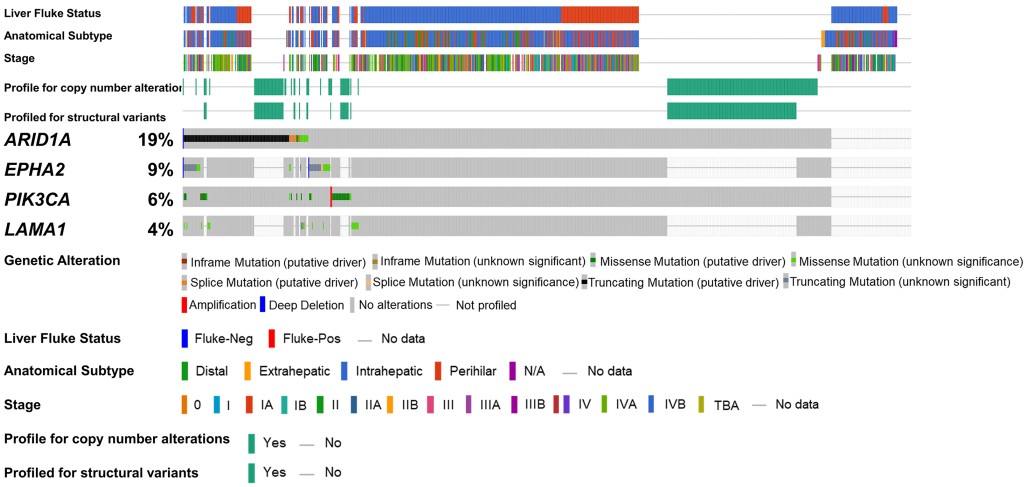

**Figure 1** Genetic alterations of the *ARID1A* mutations with alterations of *EPHA2*, *PIK3CA* and *LAMA1*. The oncoplot shows a co-occurrence pattern of *ARID1A* mutations with *EPHA2*, *PIK3CA* and *LAMA1* genes in 795 CCA patients.

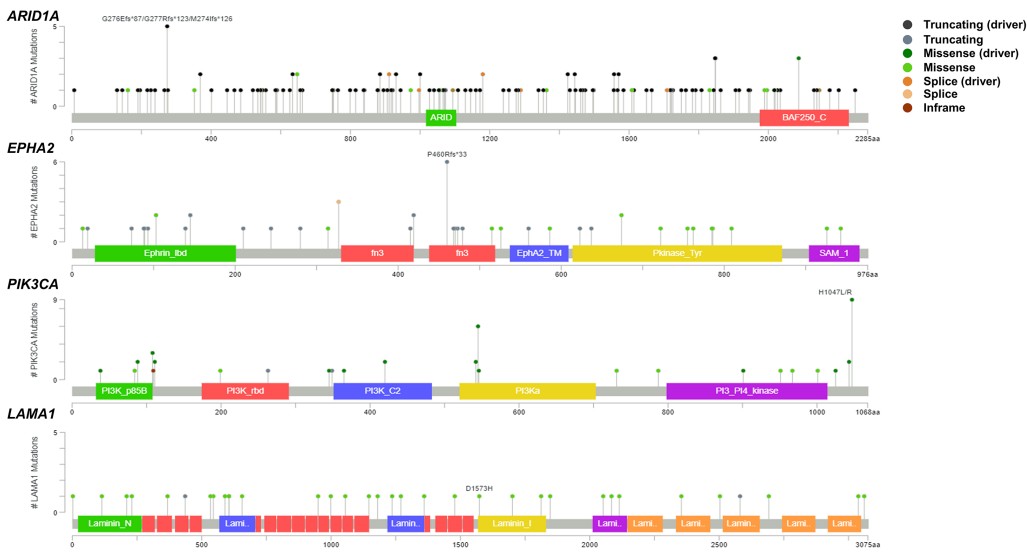

**Figure 2** Variant distribution of *ARID1A*, *EPHA2*, *PIK3CA* and *LAMA1* genes in 795 CCA patients. Lollipop plots showing the distribution of ARID1A, EPHA2, PIK3CA and LAMA1 genes in CCA patients. The most frequent variant alterations are annotated as on top of the plots.

and *ARID1A-LAMA1* mutations might be a result of limited power of sample size ($n = 9$ and $n = 6$, respectively).

## Loss of ARID1A expression in CCA cell lines led to increased sensitivity towards the AKT-inhibitor MK-2206

Several studies suggest the interdependency between loss of ARID1A protein expression and PI3K/AKT pathway activation which may also be more vulnerable to its inhibition

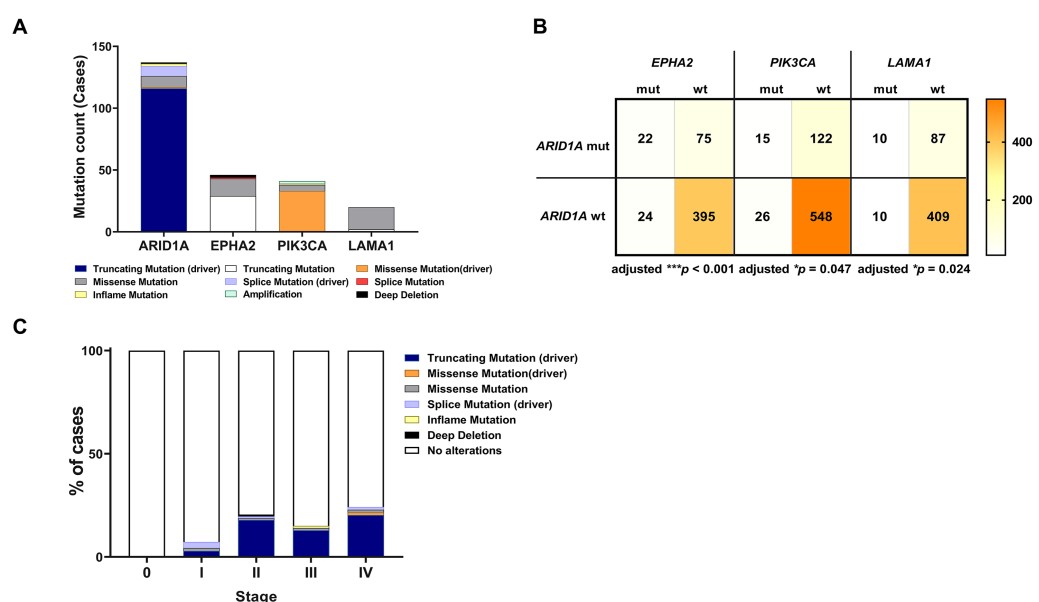

**Figure 3 Frequencies of the ARID1A alterations and its co-occurrence with alterations of *EPHA2, PIK3CA, LAMA1* and CCA staging.** (A) Frequencies of *ARID1A, EPHA2, PIK3CA,* and *LAMA1* mutations in the cBioPortal database. (B) The co-alteration incidence for *ARID1A, EPHA2, PIK3CA,* and *LAMA1* mutation. (C) Frequencies of *ARID1A* gene mutations changes across different stages of CCA. adjusted * $p < 0.05$, adjusted ** $p < 0.01$, adjusted *** $p < 0.001$ was considered statistically significant (mut: mutant, wt: wild-type).

(*Samartzis et al., 2014*; *Zhang et al., 2016*; *Lee et al., 2017*). We previously demonstrated that CCA tumors with *ARID1A* truncating mutations exhibited the loss or reduction of ARID1A protein expression (*Namjan et al., 2020*). However, the effect of PI3K/AKT inhibitor has not been well-defined in ARID1A-deficient CCA. The effect of MK-2206, however, has been investigated in early clinical trials of biliary tract cancers (NCT01425879) (*Ahn et al., 2015*). We therefore evaluated the effect of MK-2206 specifically in ARID1A-deficient CCA *in vitro*. To investigate sensitivity to treatment with MK-2206, we examined the effect of MK-2206 on cell viability in CCA cell lines. Among five CCA cell lines, KKU-213A, KKU-100 and HUCCT1 showed higher level of ARID1A protein expression compared to KKU-452 and KKU-055 cells (Figs. 5A–5B). Based on ARID1A protein expression, 4 CCA cell lines were selected as representative cells of ARID1A-depleted CCA cells (KKU-452 and KKU-055) and ARID1A-intact CCA cell lines (KKU-213A and KKU-100) for cell viability analysis. ARID1A-depleted CCA cells, KKU-452 and KKU-055 were found to be significantly more sensitive to treatment with MK-2206 (Fig. 5C). The IC$_{50}$ values of KKU-452 and KKU-055 cells were $69 \pm 13$ μM and $85 \pm 6$ μM, 24 h, respectively, while the IC$_{50}$ values of KKU-213A and KKU-100 cells were $152 \pm 17$ μM, and $120 \pm 9$ μM, 24 h, respectively ($p = 0.016$, Kruskal–Wallis test, Fig. 5D and Table S3). These results show that ARID1A-deficient CCA cells are vulnerable to MK-2206.

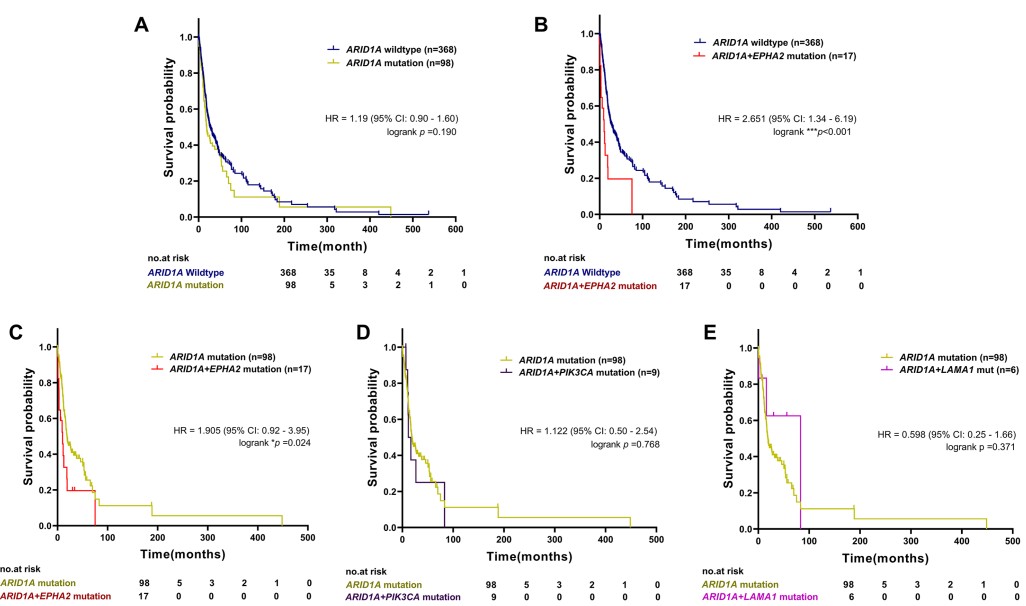

**Figure 4** **The prognostic value of ARID1A mutations and co-existent mutations with *EPHA2, PIK3CA*, and *LAMA1*.** Kaplan–Meier survival analysis of overall survival in four selected studies cohort. Data were retrieved from the cBioPortal database. *$p < 0.05$, **$p < 0.01$, ***$p < 0.001$ was considered statistically significant.

### *ARID1A* knockdown increased sensitivity to treatment with MK-2206

To confirm the effect of MK-2206 in *ARID1A*-deficient CCA cells, *ARID1A*-knockdown KKU-213A and HUCCT1 cell lines were used to investigate cell viability after the treatment with MK-2206. *ARID1A*-knockdown KKU-213A and HUCCT1 cell lines were treated with 2.5–50 μM of the MK-2206, for 24, 48 and 72 h, followed by cell viability detection. Decreased expression of ARID1A in *ARID1A*-knockdown cell lines was confirmed at mRNA and protein levels (Figs. 6A–6B). *ARID1A*-knockdown KKU-213A and HUCCT1 cell lines showed higher sensitivity towards treatment with MK-2206 when compared to the non-targeted shRNA control cells (Figs. 6C–6D). The $IC_{50}$ values of *ARID1A*-knockdown KKU-213A cells were significantly decreased (shARID1A#1 = $24 \pm 5$ μM and shARID1A#2 = $24 \pm 7$ μM, 24 h, $p = 0.016$, Fisher's LSD test) when compared to the control ($IC_{50}$ = $41 \pm 8$ μM, 24 h, Fig. 6E and Table S3). Likewise, the $IC_{50}$ values of *ARID1A*-knockdown HUCCT1 cells significantly decreased (shARID1A#1 = $21 \pm 2$ μM and shARID1A#2 = $19 \pm 2$ μM, 24 h, $p = 0.027$ and 0.013, respectively, Fisher's LSD test) when compared to the control ($IC_{50}$ = $33 \pm 8$ μM, 24 h, Fig. 6F and Table S3).

### AKT inhibition induced apoptosis in *ARID1A*-knockdown CCA cell lines and decreased phosphorylation of AKT

We subsequently investigated whether inhibition of AKT leads to increased apoptosis in *ARID1A*-knockdown cells. Treatment with MK-2206 at designated concentrations induced apoptosis in *ARID1A*-knockdown KKU-213A cell lines (Figs. 7A–7B). Flow cytometry confirmed that MK-2206 (30 μM) significantly induced apoptosis in *ARID1A*-knockdown

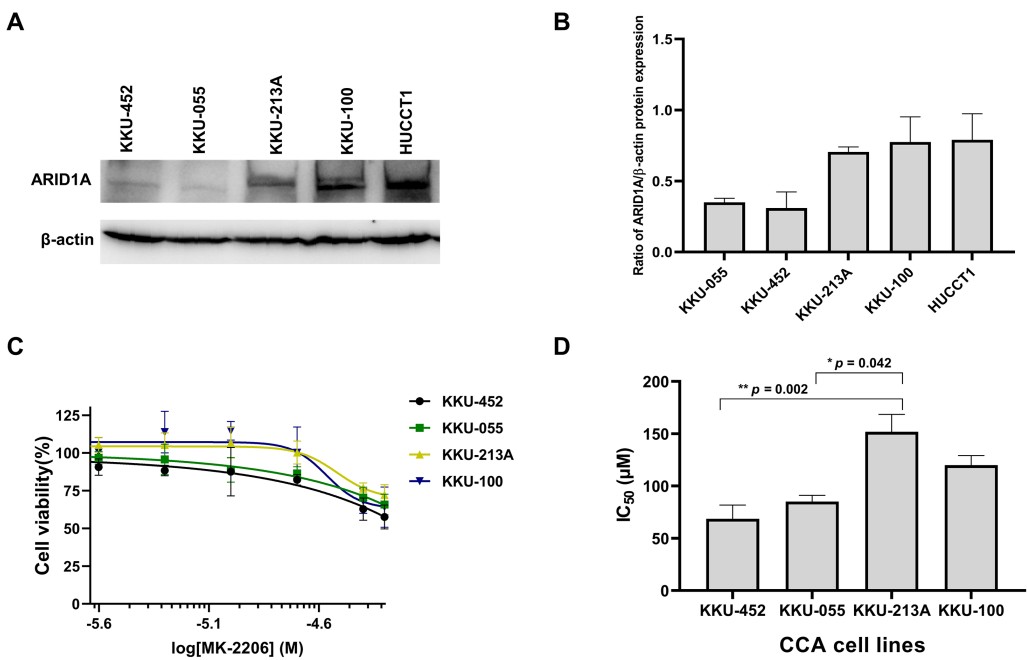

**Figure 5   Loss of ARID1A expression leads to increased sensitivity towards MK-2206 in ARID1A-deficient CCA cell lines.**  (A–B) Western blot for the screening of ARID1A expression in five CCA cell lines. (C) The effect of MK-2206 (0–50 μM) on cell viability in CCA cell lines. (D) ARID1A-depleted CCA cell lines (KKU-452 and KKU-055) were more sensitive to the treatment with MK-2206 when compared to KKU-213A and KKU-100 cells. Cell viability was measured by a MTT assay after 24 h of treatment. $*p = 0.042$, $**p = 0.002$, pairwise comparisons.

KKU-213A cell lines compared to the control (68.2% versus 44.9%, respectively, $p = 0.004$). Even though, *ARID1A*-silencing in CCA cell lines minimally increased phosphorylation of AKT at Ser-473 (pAKT$^{S473}$), treatment with MK-2206 (30 μM) significantly reduced pAKT$^{S473}$ level in *ARID1A*-knockdown KKU-213A and HUCCT-1 cell lines compared to the non-targeting control shRNA (Figs. 8A–8C). Notably, pAKT$^{S473}$ levels were significantly reduced in *ARID1A*-knockdown KKU-213A and HUCCT1 cell lines after the treatment with MK-2206 compared to control shRNA (Fig. 8C). Likewise, treatment with MK-2206 reduced mTOR protein expression in *ARID1A*-knockdown KKU-213A and HUCCT1 cells compared to the non-targeting control shRNA (Fig. 8D). Consistently, MK-2206 induced apoptosis in *ARID1A*-knockdown cell lines through the increasing of the Bax/Bcl-2 ratio (Figs. 8A–8B and Fig. S2).

## DISCUSSION

CCA is an aggressive malignancy having increased incidence globally with a high mortality rate (*Banales et al., 2020*). Most CCA patients are diagnosed in the advanced metastatic stage of the disease, resulting in poor survival and poor outcome of the local and systemic therapies (*Banales et al., 2020*). Recent comprehensive genomic profiling of CCA has revealed potential molecular targets and opened new horizons for tailored treatment for
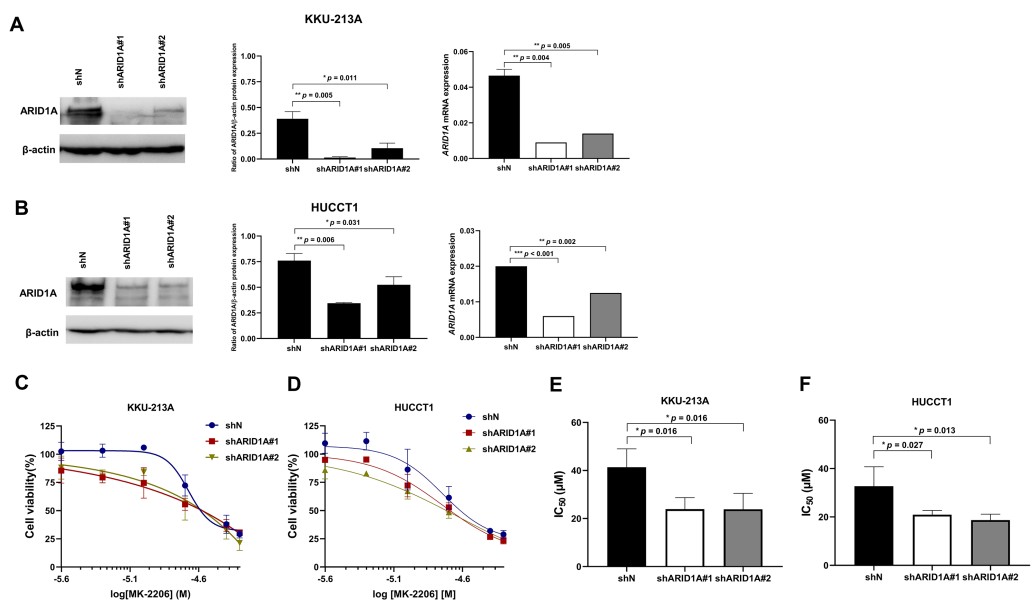

**Figure 6** ***ARID1A*-knockdown CCA cell lines show increased sensitivity towards MK-2206.** (A–B) ARID1A protein and mRNA expression were decreased in *ARID1A*-knockdown CCA cell lines (A: KKU-213A, B: HUCCT1) using a different shRNA sequence for *ARID1A* (shARID1A#1 and shARID1A#2) compared to non-targeting shRNA control cells (shN). (C–D) The effect of MK-2206 (0-50 ÂṭM) on cell viability in *ARID1A*-knockdown CCA cell lines (C: KKU-213A, D: HUCCT1). Cells were treated with 2.5–50 µM of MK-2206 for 24 h and cell viability was measured using MTT assay. (E–F) The IC$_{50}$ values of *ARID1A*-knockdown CCA cell lines (E: KKU-213A, F: HUCCT1) decreased compared to the control (shN). *$p < 0.05$, **$p < 0.01$, ***$p < 0.001$ was considered statistically significant.

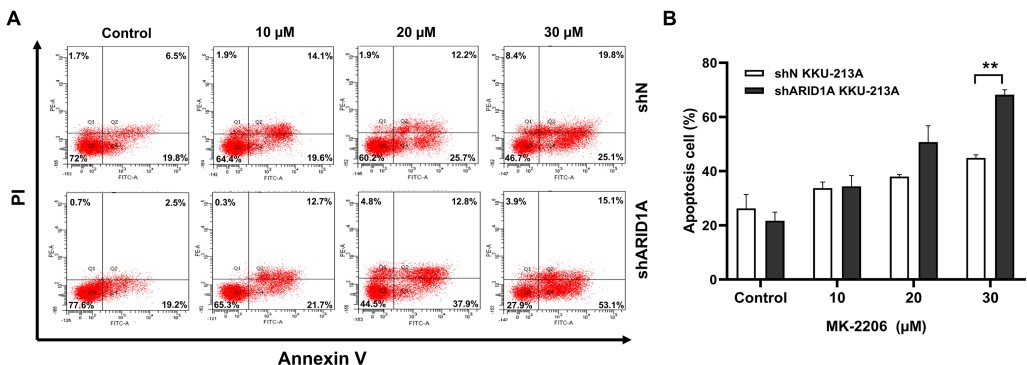

**Figure 7** **MK-2206 induces apoptosis in *ARID1A*-knockdown CCA cell lines.** (A) Flow cytometry shows that treatment with MK-2206 (24 h) led to increased apoptosis in *ARID1A*-knockdown KKU-213A cell lines (shARID1A) as compared with non-targeted shRNA control cells (shN). (B) The number of apoptotic cells was expressed as % of total cell number. **$p < 0.01$; unpaired *t*-test.

CCA. Among the chromatin remodeling genes, *ARID1A* shows one of the highest mutation rates across different cancer types in people, and it is one of the most frequently inactivated genes in CCA (*Chan-on et al., 2013*; *Jusakul et al., 2017*). Many reports suggest that ARID1A plays a tumor suppressive role in various cancers. These findings have increased interest

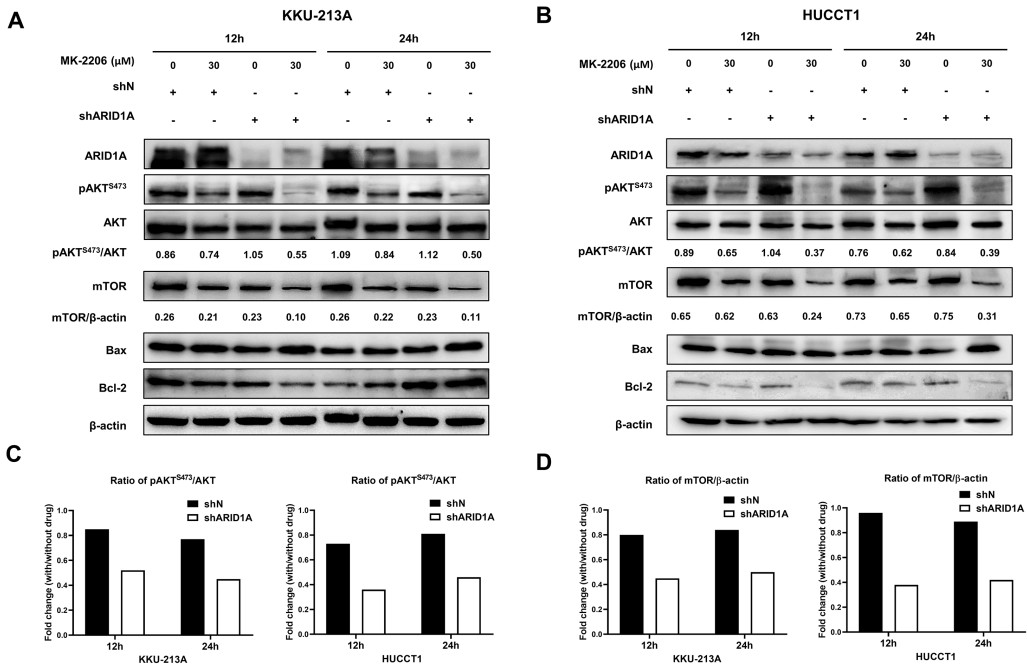

**Figure 8** **AKT inhibition decreases phosphorylation of AKT, mTOR and induces apoptosis in
*ARID1A*-knockdown CCA cell lines.** (A–B) Treatment with 30 µM MK-2206 led to decrease pAKTS473,
mTOR expression and increase Bax/Bcl-2 ratio in *ARID1A*-knockdown CCA cell lines (A; KKU-213A, B;
HUCCT1) compared to non-targeted shRNA control cell (shN). (C) Decreased ratios of pAKTS473/AKT
in *ARID1A*-knockdown CCA cell lines were observed than that of non-targeted shRNA control cells
(shN). (D) Decreased ratios of mTOR/β-actin in *ARID1A*-knockdown CCA cell lines were observed than
that of non-targeted shRNA control cells (shN). Densitometric quantification of protein expression in
CCA cell lines were obtained using ImageJ (version 1.53a, NIH, USA). Cells were treated with MK-2206
for 12 hours (12 h) or 24 h (24 h) and 0.3% DMSO was used as the control.

in developing targeted therapies that take advantage of *ARID1A* mutations. Interestingly,
*ARID1A* alterations often co-exist with genetic alterations that lead to activation of the
PI3K/AKT pathway (*Bitler, Fatkhutdinov & Zhang, 2015*). There is evidence indicating
that *ARID1A*-mutated cancers may also be vulnerable to therapeutic intervention by
targeting the PI3K/AKT pathway (*Samartzis et al., 2014*). Although, *ARID1A* inactivation
and alterations of the PI3K/AKT pathway frequently occur in CCA, the synthetic lethality
by targeting the PI3K/AKT pathway in *ARID1A*-deficient CCA has not been studied.
Herein, we demonstrated *ARID1A* mutations and its co-occurrence with alterations of
the PI3K/AKT pathway in CCA. To the best of our knowledge, this is the first time that a
synthetic lethality has been shown between ARID1A deficiency and the inhibition of the
PI3K/AKT pathway *in vitro* of CCA. Furthermore, we found that depletion of ARID1A
considerably increased sensitivity toward AKT inhibition in CCA cell lines.

Firstly, we investigated the association between *ARID1A* mutations and activation of
PI3K/AKT pathway. The PI3K/AKT pathway activation could be a result of receptor tyrosine
kinases activation or somatic mutations in specific components of the signaling pathway
such as *PTEN*, *PIK3CA*, and AKT isoforms (*Shukla & Mukherjee, 2018*). We explored

gene alterations of *ARID1A* and genes in PI3K/AKT pathway in 6 studies using the online resource cBioPortal Web. Our results indicated that *ARID1A* mutations were associated with somatic mutations of *EPHA2*, *PIK3CA*, and *LAMA1*. EPHA2, a member of the tyrosine kinase family, has been found to be frequently mutated in intrahepatic CCA (ICC). Of note, *in vitro* and *in vivo* experiments revealed that *EPHA2* mutations led to ligand-independent phosphorylation of Ser[897] and were associated with lymph node metastasis of ICC (*Sheng et al., 2019*). Additionally, *ARID1A* and *EPHA2* mutations were associated with lymph node metastasis of ICC (*Sheng et al., 2019*). In the present study, we found 82% (18/22) of *EPHA2* mutant tumors co-occurred with *ARID1A* truncating mutations, suggesting an interdependency of ARID1A and EPHA2 pathways. Interestingly, patients with *ARID1A-EPHA2* mutations were found to have a shorter overall survival than patients without *ARID1A* mutations. Additionally, we also found coexistent of *ARID1A-PIK3CA* mutations in this cohort. Coexistent *ARID1A-PIK3CA* mutations promotes tumorigenesis has been shown in several types of cancer (*Chandler et al., 2015*; *Takeda et al., 2016*; *Wilson et al., 2019*). *PIK3CA* mutations lead to dysregulation of the PI3K/AKT pathway (*Arcaro & Guerreiro, 2007*). Moreover, the H1047R/L *PIK3CA* mutations exhibited increased kinase activation and resulted in increased sensitivity to the ATP-competitive inhibitor (*Mankoo, Sukumar & Karchin, 2009*). Combination of inactivation of *ARID1A* with activation of *PIK3CA* activates the development of ovarian endometrioid carcinoma (*Wilson et al., 2019*). In ovarian clear-cell carcinomas, 40% of tumors harbor *PIK3CA* somatic mutations and the majority of these were *ARID1A*-deficient tumors (*Yamamoto et al., 2012*). Here, we found 30% (10/33) of CCA with *PIK3CA* driver missense (E545K, H1047L, R88Q, R108H, M1043I, and K111E) mutations harbored *ARID1A*-truncated mutations. This evidence suggests a synergistic mode of *ARID1A* mutations and *PIK3CA* activation, which resulted in the activation of the PI3K/AKT pathway. Importantly, our results have shown for the first time the association between mutations in *ARID1A* and *LAMA1*. LAMA1 is a subunit of laminins family. Laminins are the main component of the basement membrane, and they can promote tumor growth and metastasis (*Patarroyo, Tryggvason & Virtanen, 2002*; *Engbring & Kleinman, 2003*). In clear cell renal cell carcinoma, *LAMA1* is one of the markers associated with early metastatic cancer (*Yang et al., 2017*). We have shown that 60% (6/10) of *LAMA1*-mutated CCA co-occurred with truncating mutations of *ARID1A*. Hence, tumors with *ARID1A* deficiency may depend more on the activation of the PI3K/AKT pathway.

Currently, molecular therapies targeting the PI3K/AKT signaling pathway are under investigation in a clinical trial for several malignancies (NCT01307631 and NCT01277757). The effectiveness of AKT inhibitor (MK-2206) in inducing apoptosis was reported as a monotherapy in CCA cell lines *in vitro* (*Wilson et al., 2015*). A phase II clinical trial also evaluated the efficacy of MK-2206 on biliary tract cancers (NCT01425879). However, the clinical efficacy has been limited, to date, possibly because of the lack of appropriate patient selection based on a reliable biomarker(s). Interestingly, *ARID1A*-deficient cancers are more sensitive to PI3K/AKT inhibitors (*Samartzis et al., 2014*; *Takeda et al., 2016*; *Zhang et al., 2016*; *Lee et al., 2017*). Here, we demonstrated that *ARID1A*-deficient CCA cells show increased sensitivity to treatment with AKT inhibitor *in vitro*. *Samartzis et al. (2014)*

have indicated that *ARID1A*-deficient breast carcinoma cell lines and human primary lung fibroblasts increased sensitivity to AKT-inhibitors MK-2206, perifosine and PI3K-inhibitor buparlisib (*Samartzis et al., 2014*). Moreover, *Yang et al. (2018a)* and *Yang et al. (2018b)* showed that PI3K/AKT inhibitors (LY294002 and MK-2206) could alleviate radioresistance through the induction of apoptosis and weakening DNA damage repair in *ARID1A* mutant pancreatic cancer cells (*Yang et al., 2018a*). *Lee et al. (2017)* showed that *ARID1A*- deficient gastric cell lines were more vulnerable to AKT inhibitor GSK690693 (*Lee et al., 2017*). Likewise, we have shown that MK-2206 targeted pAKT$^{S473}$ downregulated CCA cell proliferation and induced apoptosis, conferred by ARID1A depletion. This suggests a synthetic lethal interaction between loss of ARID1A and inhibition of the PI3K/AKT pathway. In contrast to ovarian clear cell and endometrioid carcinomas (*Samartzis et al., 2014*), the effect of *ARID1A* knockdown on AKT inhibition was relatively small in CCA *in vitro* which could be a result of weak activation of AKT after *ARID1A* knockdown. Additionally, the IC$_{50}$ values reported in this study were higher than those other *in vitro* models (*Samartzis et al., 2014*; *Lee et al., 2017*; *Ewald et al., 2013*; *Wilson et al., 2015*). A previous study demonstrated that the antiproliferative and AKT inhibition effects of MK-2206 were varied among cell lines with different genetic background (*Hirai et al., 2010*), suggesting that other mechanisms responsible for the synthetic lethality of ARID1A inactivation and AKT inhibition remain to be elucidated.

There were some limitations in our study. Although our data provide in silico information regarding the association between *ARID1A* mutations and PI3K/AKT pathway, its exact mechanism and function in human cancer cells has yet to be fully elucidated. We also acknowledge that a limitation of our study is the lack of *in vivo* experiments. Future work should investigate MK-2206 properties *in vivo* and center on developing more effective combination therapies to improve treatment efficacy. Further assessment of the mechanism of action could shed light on the clinical utility of using AKT inhibitors to treat CCA patients harboring *ARID1A* alterations.

## CONCLUSIONS

Our results have demonstrated that depletion of ARID1A leads to a significantly increased sensitivity towards AKT-inhibition in CCA cells *in vitro*. Additionally, our results have shown the co-occurrence of genetic alterations of *ARID1A* with the PI3K/AKT pathway in CCA tumors. The findings suggest a synthetic lethal interaction between the loss of ARID1A and the inhibition of the PI3K/AKT pathway. Furthermore, results from our study provide a sound basis and a unique opportunity for predicting favorable treatment responses to small molecule inhibitors of the PI3K/AKT pathway on *ARID1A*-mutated CCA which can improve treatment outcomes.

## ACKNOWLEDGEMENTS

We would like to thank the patients and Cholangiocarcinoma Research Institute, Khon Kaen University, Thailand. We would like to acknowledge Professor Ross Hector Andrews, for editing the manuscript via Publication Clinic KKU, Thailand.

### Funding

This work was financially supported by Centre for Research and Development of Medical Diagnostic Laboratories (CMDL), Faculty of Associated Medical Sciences, Khon Kaen University, and the Thailand Research Fund (MRG6280073). The funders had no role in study design, data collection and analysis, decision to publish, or preparation of the manuscript.

### Grant Disclosures

The following grant information was disclosed by the authors:
Centre for Research and Development of Medical Diagnostic Laboratories (CMDL).
Faculty of Associated Medical Sciences.
Khon Kaen University.
The Thailand Research Fund:  MRG6280073.

### Competing Interests

The authors declare there are no competing interests.

### Author Contributions

- Supharada Tessiri conceived and designed the experiments, performed the experiments, analyzed the data, prepared figures and/or tables, authored or reviewed drafts of the paper, and approved the final draft.
- Anchalee Techasen and Sarinya Kongpetch conceived and designed the experiments, analyzed the data, authored or reviewed drafts of the paper, and approved the final draft.
- Achira Namjan analyzed the data, prepared figures and/or tables, and approved the final draft.
- Watcharin Loilome, Waraporn Chan-on and Raynoo Thanan conceived and designed the experiments, authored or reviewed drafts of the paper, and approved the final draft.
- Apinya Jusakul conceived and designed the experiments, analyzed the data, prepared figures and/or tables, authored or reviewed drafts of the paper, and approved the final draft.

### Data Availability

The raw data is available in the Supplementary Files. The raw western blot images are available at figshare: Tessiri, Supharada; Jusakul, Apinya (2021): Raw data for Therapeutic Targeting of ARID1A and PI3K/AKT Pathway Alterations in Cholangiocarcinoma.zip. figshare. Figure. https://doi.org/10.6084/m9.figshare.16974859.v2.

### Supplemental Information

Supplemental information for this article can be found online at http://dx.doi.org/10.7717/peerj.12750#supplemental-information.

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
