# Peer review of "Therapeutic targeting of ARID1A and PI3K/AKT pathway alterations in cholangiocarcinoma"

_PeerJ, doi:10.7717/peerj.12750_

## Round 0.1 · original submission · Major Revisions

It is imperative that the revised manuscript satisfactorily addressed the concerns identified by all reviewers including what seems to be a significant flaw about the experimental design raised by reviewer #1.

Reviewer 1 ·

Basic reporting

No comment

Experimental design

This study is highly flawed in many respects, the most obvious one being the incredibly high doses of MK-2206 used (10-150uM, and 30uM for the western blots). This is 10-1000 times higher than most publications. At this dose, there are too many off-target effects to be believable. Other issues are lack of clear PI3K pathway activation after shARID1A, weak "sensitization" to MK-2206 after shARID1A (Fig. 5C-D), and lack of connection between the experiments and the EPHA2 loss finding. Additionally EPHA2 is not really a canonical PI3K member.

Validity of the findings

Conclusions are based on faulty study design, see above.

Additional comments

No comment.

·

Basic reporting

In this paper, with the analysis on large-scale patient data, the authors reported the correlation of ARID1A mutation and PI3K/AKT pathway in cholangiocarcinoma (CCA), which would be useful for understanding the current therapeutic outcome and developing future targeted therapy. This is an innovative and interesting study. From the perspective of academic criticism, several technical concerns need to be addressed to further improve the quality of this manuscript, as appended below.

Missing citation: Line 74, 76, 78, 85, 93, 105, 259, 261, 263.

Gramma issue and ambiguous sentence: Line 74, 81, 93, 110-113, 113-115.

The introduction is a stack of information rather than a logical story. The cancer genetics and molecular background of ARID1A and PI3K/AKT pathway need to be sorted and summarized. As the author is trying to tell a therapeutical story, the state-of-art CCA targeted therapy should be the focused point of the introduction. Please re-organize the language and information accordingly.

Experimental design

Fig. 1 and Fig. 2A are hard to read due to the resolution and font size. It is reasonable to truncate or split up the figure and put those genes that were not highlighted in the story in a supplementary figure.

A gene map in the main figure or a detail description about the mutation hotspot and common mutation types of ARID1A would be helpful to better understand the story.

The gene expression profile after shRNA knockdown needs to be shown.

The quantitative analysis of the western-blot results needs to be included as part of the Fig 4A, 5A, 5B.

Validity of the findings

The insignificant shortening survival rate in the ARID1A-PIK3CA mutations and ARID1A-LAMA1 mutations need to be explained after result description.

As mentioned in the introduction, ARID1A expression has been shown to have dynamic regulation on cancer progression in different types of cancer (Sun, Xuxu, et al. "Arid1a has context-dependent oncogenic and tumor suppressor functions in liver cancer." Cancer cell 32.5 (2017): 574-589.). It would be interesting to look into the ARID1A mutation and expression pattern at different stage of CCA with the existing patient dataset and visualize the data in a stage-dependent way.

The mutation status of ARID1A has also been suggested to influence the outcome of immunotherapy (Li, Jing, et al. "Epigenetic driver mutations in ARID1A shape cancer immune phenotype and immunotherapy." The Journal of clinical investigation 130.5 (2020): 2712-2726.). It would also be interesting to explore the ARID1A mutation-related therapeutic outcome in the CCA patient dataset.

Reviewer 3 ·

Basic reporting

Supharada et al analyzed the public database of cholangiocarcinoma (CCA) patients and identified genetic alterations in the ARID1A gene. The authors subsequently performed functional studies by knockdown and overexpression of ARID1A in cell lines to show increased sensitivity to AKT inhibitor (MK-2206). The authors concluded that ARID1A deficient CCA tumors are dependent on PI3K/AKT pathway and may be more vulnerable to AKT inhibition.

Experimental design

Overall, the study is well designed. The authors performed studies using multiple CCA cell lines which is ideal for robust research findings.

Validity of the findings

Results:
1- Fig 4A and B: Please provide full western blot images in the supplementary figures. Additionally, please include dose-response curves for the cell lines instead of reporting IC50 values.
2- Fig 5b: The western blot for ARID1A doesn’t provide sufficient evidence for ARID1A knockdown. Please include cleaner blots.
3- Fig 5c-f: The effect of ARID1A knockdown on AKT inhibition is minimal in the dose-response curves. Please explain why the cells were treated for only 24h with MK-2206. It may be better to treat cells for a longer time to see a more robust effect.
4- Fig 6: The authors report significantly higher apoptotic cells at 30 uM drug concentration. Is such high micromolar concentration clinically meaningful? Additionally, MK-2206 may have non-specific effects given that IC50 for AKT inhibition is less than 100nM.
5- Fig 7: Again, 30 uM drug concentration is probably not clinically meaningful. Since there is a variable amount of pAKT in shN vs shARID1A cells at baseline, the authors should report the ratio of decrease in pAKT levels with and without the drug. Overall, changes in pAKT levels with ARID1A knockdown and AKT inhibition are relatively small.

---

## Round 0.2 · accepted · Accept

Thank you very much for submitting your work to PeerJ.

·

Basic reporting

Thank the author for the efforts. The manuscript is good for publication now.

Experimental design

good

Validity of the findings

good

Reviewer 3 ·

Basic reporting

The authors have addressed all my concerns in their revised manuscript.

Experimental design

The authors have addressed all my concerns in their revised manuscript.

Validity of the findings

The authors have addressed all my concerns in their revised manuscript.